# Nutritional Status Predicts the Length of Stay and Mortality in Patients Undergoing Electrotherapy Procedures

**DOI:** 10.3390/nu16060843

**Published:** 2024-03-15

**Authors:** Joanna Popiolek-Kalisz, Tomasz Chrominski, Marcin Szczasny, Piotr Blaszczak

**Affiliations:** 1Clinical Dietetics Unit, Department of Bioanalytics, Medical University of Lublin, ul. Chodzki 7, 20-093 Lublin, Poland; 2Department of Cardiology, Cardinal Wyszynski Hospital in Lublin, al. Krasnicka 100, 20-718 Lublin, Poland

**Keywords:** nutritional status, malnutrition, electrotherapy, pacing, cardiac resynchronization therapy, arrhythmia, nutritional risk

## Abstract

(1) Background: Nutritional status is a factor that impacts the patients’ outcomes in various medical conditions including cardiovascular patients or surgical procedures. However, there is limited available information about its impact on the short-term outcomes of cardiac implantable electronic device (CIED) implantations. This study aimed to assess the relationship between nutritional status, complications, mortality risk, and length of stay at the hospital in patients undergoing CIED implantations. (2) Material and Methods: 588 patients who underwent CIED implantation in 2022 and 2023 were included in the retrospective analysis. The nutritional status assessment was performed using NRS 2002 and BMI. The implanted devices were single-chamber pacemakers (*n* = 82), dual-chamber pacemakers (*n* = 329), one-chamber ICDs (*n* = 83), dual-chamber ICDs (*n* = 19), CRT-P (*n* = 19), and CRT-D (*n* = 56). (3) Results: The regression analysis showed that the NRS 2002 score predicted the length of stay (age-adjusted: β = 1.02, *p* = 0.001) among the CIED-implanted patients. The CRT-D subgroup was particularly responsible for this relationship (β = 4.05, *p* = 0.003 after age adjustment). The analysis also revealed significant differences between the NRS 2002 score in the in-hospital death subgroups (1.75 ± 1.00 points for deaths vs. 1.00 ± 1.00 points for survivors; *p* = 0.04). There were no significant differences in nutritional status parameters regarding early complications subgroups. (4) Conclusions: This study showed that nutritional risk assessed with NRS 2002 is a predictor of length of stay (particularly for CRT-D) and mortality among patients undergoing CIED implantations. The results of the analysis point out the impact of patients’ nutritional status on short-term outcomes of CIED implantations, particularly in CRT-D implants where 1 NRS 2002 point was a predictor of a mean 4.05 days (77.2%) longer hospitalization.

## 1. Introduction

Nutritional status is a factor that can potentially impact the patients’ outcomes in various medical conditions. According to the definition by the European Society for Clinical Nutrition and Metabolism (ESPEN), malnutrition is a condition that leads to physical and mental activity impairment and can affect the treatment outcome [1]. That is why nutritional status assessment is an important but often neglected aspect of clinical performance in terms of treatment efficacy and risk, especially in patients undergoing invasive treatment such as surgical patients. Thus, proper malnutrition detection is important for the success of the introduced therapy. As already mentioned, the impact of nutritional status on prognosis after surgical procedures is widely recognized [1,2]. Among cardiovascular patients, this relationship was proven in outcomes after cardiac surgery, and better nutritional status improves patients’ survival in coronary events and after cardiac surgery [3,4,5,6,7,8]. On the other hand, the obesity paradox is a phenomenon based on the lower acute mortality risk of obese patients compared to malnourished patients; however, recent studies that included more detailed nutritional parameters do not confirm this relationship [9,10,11].

Nutritional status is a significant factor that impacts the course of the disease, including prognosis and its treatment outcome also among cardiovascular patients without surgical treatment [1,12]. In numerous studies, Czapla et al. reported that nutritional status is correlated with the length of hospitalization in patients with hypertension and atrial fibrillation [13,14]. Moreover, nutritional status was also associated with mortality in patients with acute heart failure and coronary artery disease [15,16,17,18,19]. Furthermore, nutritional status is also associated with a better prognosis in chronic heart failure [20] and symptom severity in coronary heart disease [21]. Nutritional status is a factor that also impacts mortality in hospitalized elderly patients [22].

In addition, sarcopenic obesity is another condition associated with higher mortality risk [23,24,25]. It is characterized by excessive fat mass combined with reduced muscular mass; thus, it cannot be identified by body mass measurements alone [24]. Studies suggest that sarcopenic obesity can result in even worse outcomes than sarcopenia alone [24,25].

Nutritional status and nutritional risk can be assessed with various methods, e.g., descriptive scales, anthropometrical parameters such as body mass index (BMI), or body composition assessments with bioelectrical impedance analysis [26,27,28]. In clinical practice, nutritional risk and body mass are compulsory parameters for the nutritional status assessment among patients admitted to any hospital in Poland. Nutritional risk is assessed with recognized and widely available scales, such as the Subjective Global Assessment (SGA) or the Nutritional Risk Screening 2002 (NRS 2002) [29,30]. An NRS 2002 score is calculated based on the answers to a few simple questions referring to nutritional status deterioration. For this purpose, a patient is asked about changes in their BMI or changes in their food intake before admission to the hospital (0–3 points) and is assessed in terms of underlying diseases which can potentially lead to increased energy intake demands (0–3 points). Moreover, one additional point is given if the patient is over 70 years old. A score of at least 3 points is an indication that nutritional intervention is required [29]. In Poland, nutritional risk screening with NRS 2002, as already mentioned, is compulsory during admission to any hospital.

BMI is an anthropometric parameter that is calculated by dividing the body mass expressed in kilograms by the square of height expressed in meters. It is inexpensive and non-invasive. A BMI value is used to classify a patient’s nutritional status according to the World Health Organization’s recommendations [1]. A BMI value < 18.5 kg/m^2^ is classified as underweight, 18.5–24.99 kg/m^2^ is a normal BMI value, 25.0–29.99 kg/m^2^ is overweight, and ≥30 kg/m^2^ is obesity [31]. Obesity can be further graded as I grade: 30.00–34.99 kg/m^2^, II grade: 35.00–39.99 kg/m^2^, and III grade ≥ 40 kg/m^2^ [31]. According to the ESPEN guidelines, BMI is still a parameter advised and utilized for malnutrition diagnosis (the cut-off is below 18.5 kg/m^2^) [1]. On the other hand, despite its popularity, BMI is debated as an insufficient parameter to provide information on body composition including muscle or fat tissue content, which can be crucial in terms of a patient’s assessment [28].

Cardiac implantable electronic device (CIED) implantations are procedures that compared to open-heart cardiosurgical procedures can be classified as semi-invasive. They require a skin incision (approximately 5–7 cm long) along with a subcutaneous pocket formation; however, the electrodes’ placement is performed intravenously through the subclavian vein. Interestingly, with such a strong background regarding the impact of nutritional status on surgical outcomes, there is limited information available about the impact of undernutrition on electrotherapy procedures’ results or complications. There is published evidence for the role of malnutrition in long-term mortality after selected electrotherapy procedures, i.e., pacemakers, implantable cardioverter-defibrillator (ICD), or cardiac resynchronization therapy (CRT) implantations [32,33,34]. There is only one published study that showed a higher in-hospital mortality of malnourished patients undergoing pacemaker implantations [35]. There are no available publications that focus on in-hospital mortality after other types of CIED implantations. Regarding non-fatal, clinical outcomes of CIED implantations, the study by Balli et al. revealed that poorer immunonutritional status was also associated with the development of pacing-induced cardiomyopathy [36]. Better nutritional status was also related to better responses to CRT in a study by Ikeya et al. [37]. However, there have not been any papers published focused on the length of hospital stay after CIED implantation in the context of nutritional status.

The presented study aimed to assess the relationship between the nutritional status in patients facing CIED implantations, and complications, in-hospital mortality risk, or their length of stay at the hospital.

## 2. Materials and Methods

All the patients who underwent primary CIED implantation in 2022 and 2023 at the Department of Cardiology of Cardinal Wyszynski Hospital in Lublin were included in this retrospective study. The inclusion criteria were as follows: primary CIED implantation (single- or double-chamber pacemaker, ICD, CRT pacing [CRT-P], or CRT defibrillator [CRT-D]). The exclusion criteria were as follows: CIED replacement procedures, CIED procedures related to long-term complications after CIED implantation in the past, and leadless pacemaker implantations. The analysis included 588 patients (58.33% were men and 41.67% were women) who underwent CIED implantation: single-chamber pacemakers: 13.95% (82), dual-chamber pacemakers: 55.95% (*n* = 329), one-chamber ICDs: 14.12% (*n* = 83), dual-chamber ICDs: 3.23% (*n* = 19), CRT-P: 3.23% (*n* = 19), and CRT-D: 9.52% (*n* = 56).

The nutritional risk assessment was performed with NRS 2002. The procedure of the NRS 2002 score calculation and interpretation was performed in line with the recognized instructions and procedures described above.

Body mass was measured with 0.05 kg accuracy and the patient’s height was measured with 1.0 cm accuracy by a professional who admitted the patient to the hospital.

The information about procedure complications, their type, and in-hospital mortality was acquired from the internal departmental procedural complications registry.

### 2.1. Statistical Analysis

Statistical analyses were conducted with the STATA/BE 18.0 software. The variables were presented as means (±SD). The relationship between selected parameters and the length of stay at the hospital was investigated using a linear regression model. As the NRS 2002 score is age-dependent, the regression results were also age-adjusted. The linear regression was also performed in the subgroups by CIED type.

Then, the patients were divided into subgroups by complication, complication type, and mortality. The comparison between the two subgroups was conducted with the Mann–Whitney-U test. For binary outcomes (complications and mortality) in men and women, risk ratios were calculated. The age adjustments for the binary outcomes were performed with logistic regression.

The patients were also divided into 6 subgroups by WHO nutritional status class (underweight, normal weight, overweight, obese I, II, and III grade), and subgroups by separate NRS 2002 score. The comparison between the subgroups was performed with a one-way ANOVA test.

In all the above-mentioned statistical analyses, a *p*-value below 0.05 was recognized as significant.

### 2.2. Ethical Concerns

The study was approved by the local Bioethics Committee of the Medical University of Lublin on 21 September 2023 (consent no. KE-0254/206/09/2023). Due to the retrospective nature of the study, the written consent requirement was waived for the individual participants.

## 3. Results

### 3.1. Study Population

The analysis included 588 patients (58.33% were men and 41.67% were women) who underwent CIED implantation: single-chamber pacemakers: 13.95% (82), dual-chamber pacemakers: 55.95% (*n* = 329), one-chamber ICDs: 14.12% (*n* = 83), dual-chamber ICDs: 3.23% (*n* = 19), CRT-P: 3.23% (*n* = 19), and CRT-D: 9.52% (*n* = 56). The mean age was 73.97 ± 9.99 years. The mean length of stay at the hospital was 4.39 ± 4.47 days. The in-hospital mortality rate after CIED implantations was 0.68% (*n* = 4). The complication rate was 6.29% (*n* = 37), and among them pneumothorax accounted for 1.87% (*n* = 11)—29.73% of all complications, dislocation for 4.25% (*n* = 25)—67.57% of all complications, perforation for 0.85% (*n* = 5)—13.51% of all complications, massive hematoma which required revision accounted for 0.34% (*n* = 2)—5.41% of all complications, there was one case of vein obstruction, and one case of respiratory tract bleeding. The detailed study group characteristics are presented in Table 1.

### 3.2. Nutritional Status

Body mass information was available for 575 patients. The mean BMI was 28.55 ± 4.97 kg/m^2^. The mean body mass was 81.05 ± 16.78 kg. According to WHO’s BMI classification, 0.17% (*n* = 1) of patients were underweight, 23.65% (*n* = 136) had normal body mass, 40.52% (*n* = 233) were overweight, 24.54% (*n* = 141) were 1st grade obese, 8.35% (*n* = 48) were 2nd grade obese, and 2.78% (*n* = 16) were 3rd grade obese. NRS 2002 information was available for 488 patients. The mean NRS 2002 score was 0.90 ± 0.78 points. Separate NRS scores were 0 points: 28.66%, 1 point: 58.97%, 2 points: 7.22%, 3 points: 4.54%, 4 points: 0.21%, and 5 points: 0.41%.

### 3.3. Length of Stay

The mean length of stay at the hospital was 4.39 ± 4.47 days. The regression analysis revealed that NRS 2002 was a main nutritional factor of the length of stay (β = 0.78, R^2^ = 0.02, *p* = 0.003); after age adjustment, the relationship was even stronger (β = 1.02, R^2^ = 0.02, *p* = 0.001); similarly so after age and CIED type adjustments (β = 0.97, R^2^ = 0.04, *p* = 0.001); and age and heart failure adjustment (β = 0.96, R^2^ = 0.03, *p* = 0.001). The subgroup analysis by separate CIED types revealed that the CRT-D subgroup was particularly responsible for this relationship with β = 3.79, R^2^ = 0.21, *p* = 0.02, and even higher after age adjustment (β = 4.05, R^2^ = 0.18, *p* = 0.003). The subgroup analysis also showed that lower BMI in patients undergoing one-chamber ICD implantations was related to a longer length of stay (β = −0.23, R^2^ = 0.05, *p* = 0.04) without the relationship for CIED overall. The detailed results are presented in Table 2 and Table 3. Moreover, the analysis by the BMI WHO classification subgroups regarding the co-existing nutritional risk showed that the NRS 2002 score is an age-adjusted predictor of hospitalization length in overweight (β = 1.40, R^2^ = 0.06, *p* < 0.001) and grade I obesity (β = 1.50, R^2^ = 0.03, *p* = 0.01), and also in overall excessive body mass (overweight and obese) (β = 1.27, R^2^ = 0.04, *p* < 0.001) or without obesity (β = 0.96, R^2^ = 0.03, *p* = 0.01).

The multiple regression analysis revealed that the models, which along with NRS 2002 also included indications for the procedure, predicted the length of stay. The valid models were created for second-grade atrioventricular block, sick sinus syndrome, heart failure, and for secondary prevention in ICD receivers. The detailed results are presented in Table 4.

The ANOVA analysis between separate NRS 2002 scores and the WHO nutritional status subgroups revealed a significant difference between NRS 2002 score subgroups and the length of stay (*p* < 0.001), while this relationship was not present in BMI subgroups (*p* = 0.09). The locally weighted scatterplot smoothing for differences in the length of stay and NRS 2002 score is presented in Figure 1.

### 3.4. Complications

The complication rate was 6.29% (*n* = 37), and regarding separate types of complications, the complication rates were as follows: pneumothorax 1.87% (*n* = 11)—29.73% of all complications, dislocation 4.25% (*n* = 25)—67.57% of all complications, perforation 0.85% (*n* = 5)—13.51% of all complications, massive hematoma which required revision 0.34% (*n* = 2)—5.41% of all complications, there was one case of vein obstruction, and one case of respiratory tract bleeding.

The comparison between patients with and without complications did not reveal any significant differences in NRS 2002 scores, BMI, body mass, or age. The subgroup analysis of each type of complication also did not reveal any significant differences. The age or medical condition adjustments also did not reveal any impact of nutritional status factors on early complications.

The analysis of the relationship between sex and complication incidence did not reveal the impact of sex on any type of complication or complications overall. The results for the male sex were RR = 0.754, *p* = 0.37 for overall complications; RR = 1.25, *p* = 0.72 for pneumothorax; RR = 0.909, *p* = 0.81 for dislocation; RR = 1.071, *p* = 0.94 for perforation; and RR = 0.714, *p* = 0.81 for hematoma.

### 3.5. In-Hospital Mortality

The analysis revealed significant differences between the NRS 2002 scores in the in-hospital death subgroups (1.75 ± 1.00 points in deaths vs. 1.00 ± 1.00 points in survivors; *p* = 0.0384). However, the age-adjusted logistic regression showed that this relationship was not significant after age adjustment (*p* = 0.07). The addition of medical conditions into the regression model also did not confirm the impact of nutritional factors on in-hospital deaths. Such a relationship was also not revealed for BMI (*p* = 0.88), body mass (*p* = 0.88), or age (*p* = 0.28). The relationship between sex and mortality was also not significant (RR = 0.714, *p* = 0.73). The boxplot showing the differences in NRS 2002 scores between survival groups is presented in Figure 2.

## 4. Discussion

This is the first study that combined CIED hospitalization aspects of the length of stay, in-hospital mortality, and early complications with nutritional factors that are potentially reversible and treatable.

The analysis of the length of stay showed that a higher NRS 2002 value is a predictor of the length of stay. It is important that prolonged hospitalizations bring an increase in healthcare costs and pose a risk of infectious complications. The regression results showed that 1 point of the NRS 2002 score is a predictor of hospitalization prolongation by 1.02 days overall and even 4.05 days in CRT-D patients. As the mean length of stay at the hospital was 4.39 days overall and 5.24 days for CRT-D patients, it shows a 23.2% and 77.2% prolongation, respectively. Similar observations were made for different cardiovascular conditions, as malnutrition assessed with NRS 2002 is a factor of longer lengths of stay in patients with arrhythmias such as atrial fibrillation [13] and a factor of in-hospital mortality in heart failure patients [16]. However, our study is the first study that analyzed the length of stay in patients undergoing CIED implantations regardless of the indication. The subgroup analyses showed that the strongest relationship was presented for CRT-D implantations, which can be explained by the fact that the main indication for these procedures is advanced chronic heart failure. This suggestion was confirmed in the multivariate regression analysis model results. The available literature shows that nutritional status is a particularly important aspect of heart failure patients, as it impacts in-hospital mortality [16] and long-term outcomes such as rehospitalizations and mortality [38]. It is also important in ambulatory patients, as it affects their quality of life, mortality, and hospitalization rate [39]. Moreover, in the study by Fu et al., preimplantation nutritional status was also important in heart failure patients who received right ventricular pacing, as it was a predictor of rehospitalization rate [40]. However, the authors of that study did not analyze the length of hospital stay and included only right ventricular pacing patients.

A very important aspect of our study was the analysis of nutritional risk in various WHO BMI subgroups. It showed that nutritional risk in overweight and I-grade obese patients impacts their hospitalization length. It suggests a possibility of sarcopenic obesity in these particular patients, which is often an underdiagnosed condition. Sarcopenic obesity is defined as excessive fat mass accompanied by reduced muscular mass [24]. It is worth noting that this complex disorder is suggested to be associated with a higher health outcome risk than low body cell mass or high fat mass alone [23,24,25]. However, detailed body composition is needed for a full diagnosis of sarcopenic obesity [28], which is why this aspect needs further research.

In terms of early complications, the study by Jing et al. showed that poor nutritional status, high BMI, and older age were risk factors resulting in postoperative complications after pacemaker implantations [41]. Our study did not prove the relationship between nutritional status and complication rate. These discrepancies might be related to the fact the mentioned study included only 124 patients, with only 10 cases of complications where the most common complication reported was hematoma, then capsular rupture and infection. The profile of the complications observed in our center was different (e.g., no cases of early infections), which can be potentially caused by different operators’ experiences.

Our study was focused on the early complications and mortality of patients undergoing CIED implantations and did not include a long-term analysis. However, it is worth noting that the available literature shows a significantly lower response rate for CRT in the moderate or severe malnutrition groups [37]. That study included 263 patients. This similar relationship was confirmed in the study by Alvarez-Alvarez, which proved that the Controlling Nutritional Status (CONUT) score was an independent risk factor of death and left ventricular reverse remodeling after CRT implantation [42]. In long-term observations, the relationship between nutritional status and CRT response can be two-way, as the study by Yamada et al. showed that improvement in cardiac function after CRT device implantation was associated with an increase in the prognostic nutritional index after 6 months [43]. The mentioned studies suggest the role of nutritional status not only for in-hospital outcomes but also in long-term clinical response. It also indicates the potential direction for future research.

The present study showed that malnutrition impacts in-hospital mortality after CIED implantations. The analysis by Kichloo et al. confirmed our observations, as it indicated that malnutrition is one of the mortality predictors in hospitalizations involving pacemaker implantation [35]. Our study showed that a higher NRS 2002 score is associated with higher mortality among CIED-implanted patients. NRS 2002 score is age-dependent, as there is an additional point for age 70 or higher. The age-adjusted analysis showed that age diminishes the impact of nutritional risk score on mortality. On the other hand, the crude NRS 2002 score is the “naive” parameter, being as such not analyzed taking into account the context of the age of the patient in everyday clinical practice, so the crude NRS 2002 score helps with the identification of patients at a particular risk of death after CIED implantation. In terms of long-term observations, preprocedural malnutrition was associated with long-term mortality in patients who underwent pacemaker implantation [32]. A study by Ikeya et al. showed that among patients undergoing CRT implantations, the moderate or severe malnutrition group assessed with a CONUT score had a significantly higher long-term mortality [37]. Moreover, the modified Glasgow prognostic score, which is another parameter that can be used for nutritional status assessment, is a predictor of long-term heart failure hospitalization and death in patients after CRT-D implantation [34]. For ICD implantations, a low prognostic nutritional index predicted all-cause mortality during long-term follow-up in patients who were implanted with ICDs, secondary to heart failure with reduced ejection fraction [33]. It is worth noting that the prognostic nutritional index is not only associated with malnutrition but also with proinflammation as it includes total lymphocyte count. Moreover, for cardiovascular patients who underwent pacemaker implantation, the lower nutritional status assessed with the prognostic nutritional index and geriatric nutritional risk index resulted in higher cumulative event rates in long-term 4-year follow-up (heart failure hospitalization, myocardial infarction, stroke, or death from any cause) [44].

The novelty of our study is that it brings information also about other types of CIED in-hospital mortality using the tool that is already introduced as the standard of care, and compared to the CONUT score or the prognostic nutritional index, it does not require additional laboratory tests. In the present study, the subgroup analysis in terms of CRT-D specified mortality rate was not performed as there were no cases of in-hospital deaths after CRT-D implantations.

## 5. Conclusions

This study showed that nutritional risk assessed with the NRS 2002 is a predictor of the length of stay and in-hospital mortality among patients undergoing CIED implantations. This shows the important role of nutritional risk assessment that should not be neglected in everyday practice, particularly in invasive procedures such as CIED implantations.

### Limitations of the Study

This study revealed that nutritional risk is a factor that impacts the length of stay and in-hospital mortality of patients after CIED implantations. The limitation of this study is that the observation was limited to in-hospital mortality; however, this study aimed to assess the early CIED implantation complications, so future studies regarding long-term follow-up are needed. Another limitation of the study is the limited number of observed death cases, which is good information in terms of medical practice but it indicated the need for future research that includes even larger study groups. This study was a single-center study, which implies that the complication rate could be affected by factors such as the operators’ experience compared to other centers. That is why multi-center studies are also needed in this field. Moreover, this study was focused on nutritional aspects as the risk factors, so recognized prognostic scores such as the CHA_2_DS_2_-VASc Score were not used. More studies concerning new risk factors are also needed.

## Figures and Tables

**Figure 1 nutrients-16-00843-f001:**
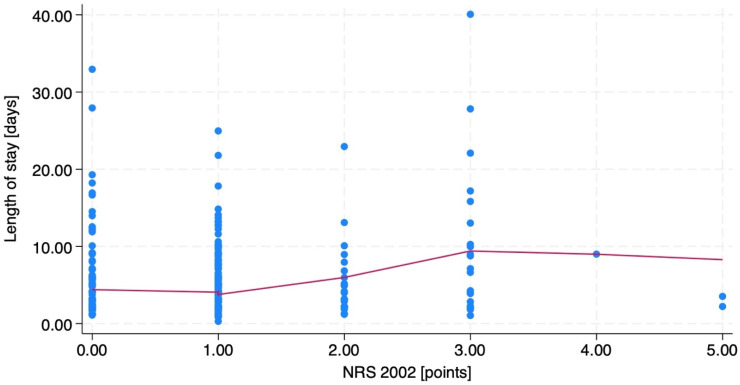
The locally weighted scatterplot smoothing for differences in the length of stay and NRS 2002 score. The blue dots are separate patients’ marks and the red line is the locally weighted scatterplot smoothing curve.

**Figure 2 nutrients-16-00843-f002:**
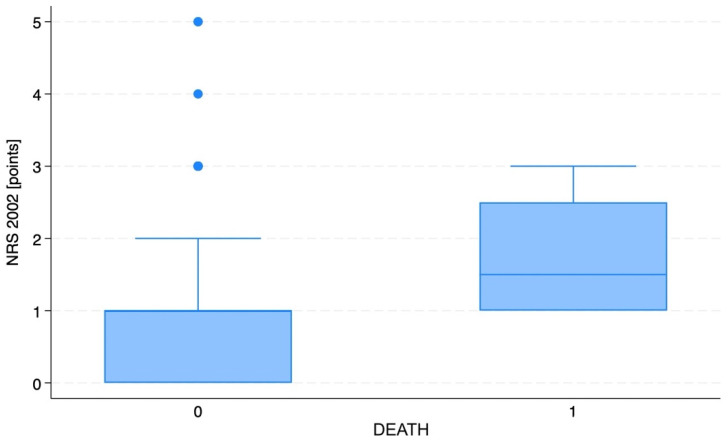
The boxplot for the differences in NRS 2002 scores between survival groups. The blue dots are extreme outliers.

**Table 1 nutrients-16-00843-t001:** The study group characteristics (mean ± SD).

	Overall	Single-Chamber Pacemakers	Dual-Chamber Pacemakers	Single-Chamber ICD	Dual-Chamber ICD	CRT-P	CRT-D	*p*
Sex [% of males]	58.33	54.88	48.33	83.13	68.42	73.68	76.79	<0.001
Age [years]	73.97 ± 9.99	79.36 ± 8.53	75.22 ± 9.22	66.70 ± 9.12	71.21 ± 11.97	74.06 ± 13.31	70.43 ± 8.72	<0.001
Length of stay [days]	4.39 ± 4.47	4.66 ± 5.41	3.76 ± 2.99	4.70 ± 4.99	7.04 ± 6.84	7.56 ± 6.87	5.24 ± 6.31	<0.001
Complications rate [%]	6.29	6.10	5.47	3.61	10.53	0	12.50	<0.001
Mortality rate [%]	0.68	1.22	0.61	0	0	5.26	0	<0.001
BMI [kg/m^2^]	28.55 ± 4.97	28.05 ± 5.35	28.91 ± 4.93	27.96 ± 5.01	26.25 ± 3.87	30.56 ± 5.01	28.17 ± 4.64	0.67
Body mass [kg]	81.05 ± 16.78	78.38 ± 17.25	80.37 ± 16.59	83.80 ± 17.68	75.66 ± 12.70	92.61 ± 17.29	82.85 ± 15.17	0.56
NRS 2002 [points]	0.90 ± 0.78	1.13 ± 0.72	0.91 ± 0.71	0.64 ± 0.92	0.93 ± 1.16	1.00 ± 0.94	0.83 ± 0.79	0.01

BMI—body mass index, CRT-D—cardiac resynchronization therapy defibrillator, CRT-P—cardiac resynchronization therapy pacemaker, ICD—implantable cardioverter-defibrillator, NRS 2002—Nutritional Risk Screening 2002 score.

**Table 2 nutrients-16-00843-t002:** The univariate linear regression results for the impact of the selected parameters on the length of stay in CIED patients.

	Overall CIED			One-Chamber Pacemaker			Dual-Chamber Pacemaker			One-Chamber ICD			Dual-Chamber ICD			CRT-P			CRT-D		
	β	R^2^	*p*	β	R^2^	*p*	β	R^2^	*p*	β	R^2^	*p*	β	R^2^	*p*	β	R^2^	*p*	β	R^2^	*p*
NRS 2002	0.78	0.0187	0.003	0.29	0.0015	0.759	0.30	0.0048	0.248	1.01	0.0356	0.129	−0.64	0.0126	0.691	2.58	0.1164	0.180	3.79	0.2140	0.002
BMI	−0.05	0.0027	0.216	0.20	0.0443	0.066	0.002	0.000	0.960	−0.23	0.0511	0.042	−0.61	0.1184	0.149	−0.25	0.0312	0.483	−0.24	0.0304	0.207
Body mass	−0.02	0.0034	0.166	0.04	0.0196	0.224	−0.01	0.0014	0.494	−0.06	0.0428	0.064	−0.18	0.1062	0.173	−0.05	0.0159	0.618	−0.11	0.0605	0.073
Age	−0.002	0.000	0.933	−0.03	0.0028	0.647	0.02	0.0032	0.309	−0.01	0.0003	0.870	−0.07	0.0162	0.604	0.10	0.0344	0.461	0.07	0.0088	0.500

BMI—body mass index, CRT-D—cardiac resynchronization therapy defibrillator, CRT-P—cardiac resynchronization therapy pacemaker, ICD—implantable cardioverter-defibrillator, NRS 2002—Nutritional Risk Screening 2002 score.

**Table 3 nutrients-16-00843-t003:** The age-adjusted linear regression results for the impact of the selected parameters on the length of stay in CIED patients.

	Overall CIED		One-Chamber Pacemaker		Dual-Chamber Pacemaker		One-Chamber ICD		Dual-Chamber ICD		CRT-P		CRT-D	
	β	*p*	β	*p*	β	*p*	β	*p*	β	*p*	β	*p*	β	*p*
NRS 2002	1.02	0.001	0.53	0.62	0.29	0.35	1.24	0.08	−0.52	0.77	2.67	0.27	4.05	0.003
BMI	0.05	0.21	0.20	0.08	0.01	0.79	−0.24	0.04	−0.59	0.18	−0.17	0.67	−0.23	0.24
Body mass	−0.02	0.14	0.04	0.26	−0.004	0.69	−0.07	0.046	−0.17	0.21	−0.01	0.94	−0.10	0.10

BMI—body mass index, CRT-D—cardiac resynchronization therapy defibrillator, CRT-P—cardiac resynchronization therapy pacemaker, ICD—implantable cardioverter-defibrillator, NRS 2002—Nutritional Risk Screening 2002 score.

**Table 4 nutrients-16-00843-t004:** The multivariate linear regression results for the impact of the selected conditions on the length of stay in CIED patients.

	β	*p*	R^2^	*p*
NRS 2002 score	0.80	0.002	0.04	<0.001
Second-grade atrioventricular block	−1.54	0.003		
NRS 2002 score	0.77	0.003	0.03	0.001
Sick sinus syndrome	−1.26	0.01		
NRS 2002 score	0.82	0.001	0.03	<0.001
Heart failure	1.05	0.02		
NRS 2002 score *	1.42	0.01	0.09	0.003
Secondary prevention ICD *	3.08	0.03		

*—in ICD receivers.

## Data Availability

The data that support the findings of this study are available from the corresponding author upon reasonable request.

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
