# Peer review of "Nutritional Status Predicts the Length of Stay and Mortality in Patients Undergoing Electrotherapy Procedures"

_nutrients, 2024, doi:10.3390/nu16060843_

Round 1

Reviewer 1 Report

Comments and Suggestions for Authors

The Authors demonstrated that the nutritional risk assessed by NRS 2002 was a predictor of length of stay and mortality among patients undergoing CIED implantations.

The manuscript is well written and each section clearly exposed.

I congratulate the Authors for their work.

I have only one suggestion for the Authors.

In the Limitation section, the Authors could a short sentence concerning the lack of comparison of NRS 2002 with other prognostic scores that have recently been used for predicting mortality in elderly patients and in patients candidates to invasive cardiological procedures, such as CHA2DS2-Vasc Score (PMID: 35294768 and PMID: 23828875).

Author Response

The Authors demonstrated that the nutritional risk assessed by NRS 2002 was a predictor of length of stay and mortality among patients undergoing CIED implantations. The manuscript is well written and each section clearly exposed. I congratulate the Authors for their work.

I have only one suggestion for the Authors. In the Limitation section, the Authors could a short sentence concerning the lack of comparison of NRS 2002 with other prognostic scores that have recently been used for predicting mortality in elderly patients and in patients candidates to invasive cardiological procedures, such as CHA2DS2-Vasc Score 

Thank you very much for your kind feedback and appreciation of our work. We added the information in the limitations of the study section as suggested.

Reviewer 2 Report

Comments and Suggestions for Authors

In the present study Popiolek-Kalisz investigated the nutritional status in patients undergoing cardiac devices implantation. They found that nutritional status predicted early in-hospital mortality and length of stay in hospital, but not the rate of complications.

Study strengths: The study is characterized by a large sample and include patients implanted with different kind of cardiac devices.

A table reporting  anthropometric and  clinical features of patients (according to the group to which they belong) is missing. I guess there were differences in terms of age, number of drugs taken , clinical complexity  between patients of different groups.

Authors should explain why they used linear regression analysis instead of cox regression  analysis  for assessing  the relation between different variables and mortality.  

Results of regression analysis should be better presented

Is  table 1 reporting results of the multivariate analysis?  Please specify. Moreover, according to table 1 the only covariates included in the multivariate model were BMI, body mass and  age. However I guess BMI and body mass were related each other. Authors should specify whether variables were tested individually in a univariate analysis before entering a multivariate regression. Since this is a complex clinical scenario there are probably many variables affecting the outcome of these patients (underling cardiac disease, pharmacological therapy, NYHA functional class etc).  I suggest to better specify results of univariate and multivariate analysis and criteria adopted for selecting variables I n the multivariate model .

Minor observations:

Line 93: “There is only one study that…” this sentence seems to be incomplete

Discussion: overall this paragraph appear well organized, however several sentences should be improved in order to render them clearer

Comments on the Quality of English Language

It needs to be improved, particularly in the discussion pragraph

Author Response

Dear Reviewer,

Thank you very much for your time, effort, and valuable comments on our manuscript. We corrected the manuscript according to your suggestions and we believe that it deeply improved its quality. Below you can find our responses to your comments. Thank you for this opportunity.

In the present study Popiolek-Kalisz investigated the nutritional status in patients undergoing cardiac devices implantation. They found that nutritional status predicted early in-hospital mortality and length of stay in hospital, but not the rate of complications.

Study strengths: The study is characterized by a large sample and include patients implanted with different kind of cardiac devices.

Thank you very much for your kind appreciation of our work.

A table reporting  anthropometric and  clinical features of patients (according to the group to which they belong) is missing. I guess there were differences in terms of age, number of drugs taken , clinical complexity  between patients of different groups.

The Table for the study group characteristics was added (Table 1).

Authors should explain why they used linear regression analysis instead of cox regression  analysis  for assessing  the relation between different variables and mortality.  

This study was designed as a retrospective study, and the Cox regression is dedicated to application in the prospective studies. That is why we used the linear regression model for the continuous outcomes and logistic regression for the binary outcomes.

Results of regression analysis should be better presented

We corrected the presentation of the results in Table 2 to make it more clear. We also added additional tables (Tables 3 and 4) to better present the results.

Is  table 1 reporting results of the multivariate analysis?  Please specify. Moreover, according to table 1 the only covariates included in the multivariate model were BMI, body mass and  age. However I guess BMI and body mass were related each other. Authors should specify whether variables were tested individually in a univariate analysis before entering a multivariate regression. Since this is a complex clinical scenario there are probably many variables affecting the outcome of these patients (underling cardiac disease, pharmacological therapy, NYHA functional class etc).  I suggest to better specify results of univariate and multivariate analysis and criteria adopted for selecting variables I n the multivariate model .

The Table 2 (previous Table 1) results were presented more clearly to easier assess the data (the B-values were added). The presented results come from the univariate analysis. The results of the multivariate analyses were initially described in the text, we did not present it firstly in a separate table in order not to double-present the data, however now we added the result of the age-adjusted analyses as a separate table (Table 3).

The indication and underlying cardiac condition also impacted the type of implanted CIED. We wanted to focus on CIED overall and on the nutritional status parameters as the potentially reversible risk factors, which is why initially we did not analyze the indication for the procedure. However, after this thoughtful suggestion, we also performed multivariate regression analyses regarding the underlying medical condition e.g. heart failure, AVB and its grade, sick sinus syndrome, and the primary or secondary prevention in case of defibrillator implantations. Heart failure was not present is all the patients (only in 154 out of 588), which is why we focused on the parameters available in all the patients undergoing CIED implantation.

Minor observations:

Line 93: “There is only one study that…” this sentence seems to be incomplete

The sentence was corrected to make it more clear. „There is only one published study that showed a higher in-hospital mortality of malnourished patients undergoing pacemaker implantations”

Discussion: overall this paragraph appear well organized, however several sentences should be improved in order to render them clearer

The discussion section underwent extensive corrections and rearrangement to improve the readers’ reception.